

# Indigenous Australian household structure: a simple data collection tool and implications for close contact transmission of communicable diseases

Thiripura Vino[1], Gurmeet R. Singh[2,3], Belinda Davison[2], Patricia T. Campbell[4,5], Michael J. Lydeamore[4,5], Andrew Robinson[1,6,7], Jodie McVernon[5], Steven Y.C. Tong[2,8] and Nicholas Geard[9,10]

[1] School of Mathematics and Statistics, University of Melbourne, Melbourne, Victoria, Australia
[2] Menzies School of Health Research, Darwin, Northern Territory, Australia
[3] NT Medical Program, Flinders and Charles Darwin University, Darwin, Northern Territory, Australia
[4] Murdoch Children's Research Institute, The Royal Children's Hospital, Melbourne, Victoria, Australia
[5] Victorian Infectious Disease Reference Laboratory, The Royal Melbourne Hospital and The University of Melbourne, at the Peter Doherty Institute for Infection and Immunity, Melbourne, Victoria, Australia
[6] School of Biosciences, University of Melbourne, Melbourne, Victoria, Australia
[7] Centre of Excellence for Biosecurity Risk Analysis, University of Melbourne, Melbourne, Victoria, Australia
[8] Victorian Infectious Diseases Service, The Royal Melbourne Hospital and The University of Melbourne, at the Peter Doherty Institute for Infection and Immunity, Melbourne, Victoria, Australia
[9] Melbourne School of Population and Global Health, University of Melbourne, Melbourne, Victoria, Australia
[10] School of Computing and Information Sciences, University of Melbourne, Melbourne, Victoria, Australia

Corresponding author
Steven Y.C. Tong,
steven.tong@mh.org.au

## ABSTRACT

Households are an important location for the transmission of communicable diseases. Social contact between household members is typically more frequent, of greater intensity, and is more likely to involve people of different age groups than contact occurring in the general community. Understanding household structure in different populations is therefore fundamental to explaining patterns of disease transmission in these populations. Indigenous populations in Australia tend to live in larger households than non-Indigenous populations, but limited data are available on the structure of these households, and how they differ between remote and urban communities. We have developed a novel approach to the collection of household structure data, suitable for use in a variety of contexts, which provides a detailed view of age, gender, and room occupancy patterns in remote and urban Australian Indigenous households. Here we report analysis of data collected using this tool, which quantifies the extent of crowding in Indigenous households, particularly in remote areas. We use these data to generate matrices of age-specific contact rates, as used by mathematical models of infectious disease transmission. To demonstrate the impact of household structure, we use a mathematical model to simulate an influenza-like illness in different populations. Our simulations suggest that outbreaks in remote populations are likely to spread more rapidly and to a greater extent than outbreaks in non-Indigenous populations.

Subjects Anthropology, Epidemiology, Infectious Diseases, Public Health
Keywords Indigenous, Housing, Communicable diseases, Influenza, Demographics, Social contact, Aboriginal

## INTRODUCTION

Households are an important location for the transmission of communicable diseases due to the frequency, duration and strength of the interactions that occur there. Patterns of household structure in a population can influence how a disease will spread, and potentially inform how it may best be controlled. Data on household structure are therefore a valuable input into mathematical models of disease transmission used for decision making on control measures. Due to the different household structures in remote and isolated communities, it is especially important to take them into consideration in disease surveillance and control (*Laskowski et al., 2011*). Household characteristics, such as the number and ages of people resident, and the number of people per room, tend to vary across subpopulations, depending upon fertility levels, socioeconomic factors and cultural norms (*Geard et al., 2015*). Communicable diseases are a major issue in remote Indigenous populations, where respiratory infections such as influenza and skin infections such as scabies and impetigo—readily transmitted in a household context—are highly prevalent (*Flint et al., 2010*; *Trauer et al., 2011*; *Andrews et al., 2009*; *Tasani et al., 2016*).

Detailed household-level information is often not publicly available in most demographic data collection surveys including the national census. This is particularly the case in resource-limited settings where literacy levels may be low and household structures may differ markedly from the nuclear household structure typically assumed by survey designs (*Morphy, 2006*). For example, Indigenous households in Australia tend to be larger than non-Indigenous households, contain more extended family members, and may change in composition more rapidly (*Morphy, 2006*; *Morphy, 2007*). Furthermore, national censuses are resource intensive and conducted relatively infrequently. There is therefore a need for more lightweight methods that allow for rapid, repeated measurement in specific populations where literacy levels may be low. These methods would contribute in understanding the differences of household structures among Indigenous communities with more accurate data, better models for prediction of outbreaks and support decisions regarding control measures.

Here we describe a novel visually-based method for collecting data on the structure of Indigenous households and provide a descriptive analysis of data collected as part of the Aboriginal Birth Cohort (ABC) study. We compare the age-specific patterns of contact within these households to those occurring in a non-Indigenous population. Finally, we explore potential implications of observed differences in household composition for the transmission of a respiratory infection such as influenza.

## METHODS

### Study design and sample

Study design and sample information for the ABC study has been described in *Sayers et al. (2003)*. In brief, the ABC is a prospective study of 686 babies born to mothers recorded as Indigenous in the Delivery Suite Register (a representative sample of the 1,238 eligible babies), recruited at Royal Darwin Hospital (RDH) between January 1987 and March 1990. RDH is the main hospital in the Darwin Health Region, an area covering 120,000 $km^2$ of

the Northern Territory and at the time, 90% of pregnant Indigenous mothers from this region came to the RDH to deliver their babies (*Sayers & Powers, 1993*). Follow-up studies of this cohort have occurred at the mean participant age of 11 years (1998–2002) (*Sayers et al., 2003*), 18 years (2005–2007) (*Sayers et al., 2009*) and 24 years (2013–2015) (*Sayers, Mackerras & Singh, 2017*) at the participant's community of residence. Written consent was provided by participants in the ABC study. The most recent follow-up was approved by the Human Research Ethics Committee of Northern Territory Department of Health and Menzies School of Health Research, including the Aboriginal Ethical Sub-committee which has the power of veto (ABC Reference no. 2013–2022). Ethical approval was contingent on written support provided from each community's local governing bodies.

Our analyses use data obtained in the most recent follow-up when participants were aged 22–27 years. There were 459 participants seen during 2013–2015 and of these household structure data were collected for 416 participants using either an abbreviated single question questionnaire (156 respondents) or, for willing participants, a magnetic board method (260 respondents). The questionnaire asked the question "Who slept in the house last night?" to obtain the household size. This question was agreed during community consultation to be best understood and most accurately answered, unlike questions regarding household size in general.

## Household number board

In designing a simple visual tool to collect household structure data we extensively consulted with both urban and remote communities, and obtained advice on study methods. Recommendations included the need for simple explanations and data collection methods in plain English and supplemented with pictures where appropriate. The household number board was developed and piloted in direct consultation with Indigenous community members and researchers.

The household number board consists of a magnetic board depicting the house and varying sized and coloured magnets depicting occupants. De-identification occurred at point of contact, with only the participant's unique study identification number transcribed onto the top right corner of the board. The board was separated into four rooms with the provision of an extra room or verandah. The rooms were intentionally non-identified. In Indigenous communities, it is common for rooms other than bedrooms to be used as sleeping quarters. No houses had more than five rooms and we only counted occupied rooms. Different sized and colored magnets represented the following: a brown smiley face for the participant, larger blue (men) and pink (women) for adults, medium orange (boy) and purple (girl) for school aged children (5–16 years), and green (boy) and yellow (girl) for preschool (<5 years) (see Fig. 1).

The participant magnet was placed in a room on the board. Participants were then asked a series of questions including whether there was any one else sleeping in the room: another adult, man or woman? Were there any children: school aged or preschool, boys or girls? And how many of each? The appropriate magnet was then placed in the room. The number of occupied rooms was noted. This process was then repeated for each of occupied rooms.
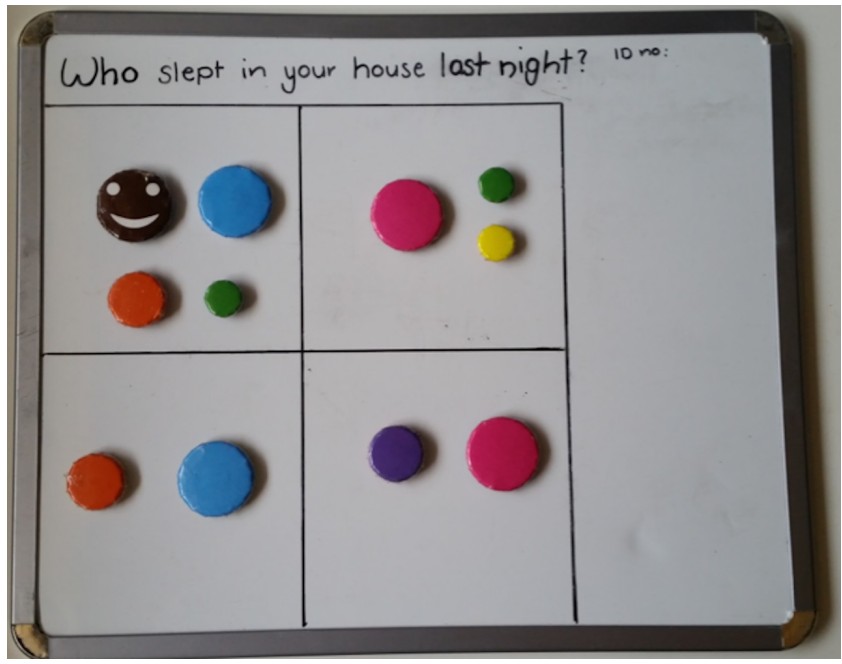

**Figure 1** **Example of completed house board.** Magnet colours identify individuals as follows: brown, participant; blue, adult man; pink, adult woman; orange, school aged boy; purple, school aged girl; green, pre-school aged boy; yellow, pre-school aged girl.

On completion, a high quality photo of the board was uploaded onto a secure computer for later analysis.

## Data preparation

Data on the number of individuals by room, age category and gender were summarised from each photo and manually entered into a spreadsheet. The sum of occupants per room was checked against household size to ensure consistency. Each household was designated as urban or remote based on Australian Bureau of Statistics Census classification.

Additional variables were constructed to summarise the total number of occupied rooms in each household, and the mean number of individuals per occupied room in each household. Town names and allocation to established shire councils (the common name for a government administrative region) were checked for accuracy and consistency.

## Analysis of household data

Summary measures were calculated for household size, and for household and room occupancy by age category and gender. Analyses were stratified by shire council, and by urban/remote status.

## Household contact matrices

Levels of household contact within and between age categories were summarised by deriving matrices of age-specific contact rates, as are commonly used to parameterise models of infectious disease transmission, as follows.

Table 1 **Household contact matrix.** Number of pre-school aged children ($b$), school-aged children ($c$) and adults ($a$).

|  | Pre-school aged | School aged | Adult |
|---|---|---|---|
| Adult | $ab$ | $ac$ | $a(a-1)/2$ |
| School aged | $cb$ | $c(c-1)/2$ |  |
| Pre-school aged | $b(b-1)/2$ |  |  |

The number of pre-school aged children ($b$), school-aged children ($c$) and adults ($a$) was extracted for each household. We assumed that each person in a household has the opportunity to come into contact with each other member of the household in any given day. The daily number of contacts between individuals within the same age category is therefore given by $x(x-1)/2$, where $x$ is the number of individuals in that age category. The daily number of contacts between individuals in different age categories is given by $xy$ where $x$ and $y$ are the respective number of people in the two age categories. 95% confidence intervals were estimated using a nonparametric bootstrap method with 1,000 bootstrap samples.

The contact matrix for an individual household, which is symmetric, is therefore given by Table 1.

Given that we also know which room the members of a household slept in, we further explored the effect of weighting the contacts between members of a household who share a room, to estimate a *weighted* number of contacts between individuals in each age category. From the perspective of disease transmission, this was intended to capture the additional risk of transmission of certain pathogens attributable to sleeping in close proximity. This would avoid underestimation of the intense and prolonged contacts (*Smieszek, 2009*). In the analyses that follow, the *room factor* reflects this weighting. A *room factor* of 1 indicates that no additional weighting was attributed to sharing a room, a *room factor* of 2 indicates that sharing a room counted twice when determining the level of contact, and so on. For example, consider a hypothetical two-room household containing two individuals ($X$ & $Y$) sleeping in one room and one individual ($Z$) sleeping in the other. In the absence of weigthing (i.e., $f = 1$) each of the three individuals would make two effective contacts per day. If we increase the weighting factor associated with sharing a room (e.g., $f = 2$), then $X$ & $Y$ would each make three effective contacts per day, while $Z$ would still make only two effective contacts per day.

Contact matrices were also stratified by shire council, and by urban/remote status. For comparison, equivalent contact matrices were derived from data collected in an urban Australian population (Melbourne; reported in *Rolls et al. (2015)*). For the purpose of designating comparable age categories, pre-school aged children were defined as those less than five years and school aged children were defined as those five to less than 16 years.

## Outbreak simulations

An age structured SEIR (Susceptible-Exposed-Infected-Recovered) model was used to simulate the outbreak of a flu-like illness in remote and urban Indigenous populations, and an urban non-Indigenous population (*Li et al., 1999*). In this model, the population


**Figure 2** **Basic SEIR model.** The four states are Susceptible ($S$), Exposed ($E$), Infected ($I$), Recovered ($R$) and the parameters are $\lambda$-rate of change from $S$ to $E$, $\sigma$-rate of change from $E$ to $I$, $\gamma$-rate of change from $I$ to $R$.

is divided into four categories as per the infection transmission process as susceptible ($S$), who can acquire infection; exposed ($E$), who have been exposed to infection and are in a latent incubation stage; infectious ($I$), who are infectious; and recovered ($R$), who are immune to the infection from natural immunity (Fig. 2).

Further, the model is divided into compartments based on the three age categories as adult, school aged and pre-school aged for the populations. The model equations for the simulation are shown in Eqs. (1)–(4). $\bar{S}, \bar{E}, \bar{I}, \bar{R}$ are vectors with values from the three age categories. $\lambda$ is the rate of change from susceptible to exposed, $\sigma$ is the rate of change from exposed to infectious and $\gamma$ is the rate of change from infected to recovered.

$$\frac{d\bar{S}}{dt} = -\bar{\lambda}\bar{S} \tag{1}$$

$$\frac{d\bar{E}}{dt} = \bar{\lambda}\bar{S} - \sigma\bar{E} \tag{2}$$

$$\frac{d\bar{I}}{dt} = \sigma\bar{E} - \gamma\bar{I} \tag{3}$$

$$\frac{d\bar{R}}{dt} = \gamma\bar{I}. \tag{4}$$

In order to calculate the transmission rate of the population, Eq. (5) was used.

$$\bar{\lambda} = q_1\bar{C}_h\bar{I} + q_2\bar{C}_c\bar{I}. \tag{5}$$

Contact matrices for household structure ($\bar{C}_h$) were calculated based on the data and the contact matrices for community structure ($\bar{C}_c$) were calculated based on the age proportions of the population derived from Australian Bureau of Statistics 2011 Census data assuming proportional mixing. When constructing community contact matrices, we assumed that an individual came into contact with 10 people per day in community settings, based on observations from (*Mossong et al., 2008*). Except for the contact matrices, the same parameters were used for each simulation. We assumed a latent period of 1.5 days, an infectious period of 1.5 days, and that probability of transmission within households ($q_1$) was twice that of transmission within community ($q_2$). We calibrated these probabilities to produce a final affected population in an urban non-Indigenous population of approximately 25% without prior immunisation or vaccination (*Ghani et al., 2010*; *Tuite et al., 2010*). The basic reproduction number $R_0$, was computed by calculating the dominant eigenvalue of the next generation matrix for each population (*Diekmann, Heesterbeek & Metz, 1990*). Both Indigenous and non-Indigenous populations were assumed to be initially susceptible, without any protection from vaccination or prior
immunity. Rather than calibrating to a specific outbreak, parameter values were chosen to illustrate the impact of different household structures on disease transmission. This age structured mathematical model was used to simulate the outbreak of an influenza-like illness to assess potential implications of the different patterns of household contact for infectious disease transmission.

## RESULTS

### Descriptive analysis

Households with data collected using the questionnaire method ($n = 156$) had a median household size of six (range one to 14) in an Indigenous remote area and a median household size of four (range one to 17) in an urban area. These results were similar to those obtained from the household magnetic board method ($n = 260$), with a median size of seven (1–23) for remote and four (1–11) for urban households. Household size data collected from the *Australian Bureau of Statistics (2011)* Census Survey also shows that more than one-third of the population has a household size of seven or more in the remote towns where ABC studies were conducted (Fig. S1). Therefore, data from the magnetic board are considered as reasonably representative of the broader Indigenous remote population and we now focus on this subset of participants. The mean age of represented participants was 25.2 years (range 23 to 27), and males and females were equally represented.

The majority of households were located in the East Arnhem shire council (41.5%, 108 households) and Victoria Daly shire council (26.5%, 69 households). Other concentrations of households were located in the Tiwi Islands (29 households), Darwin (25 households) and Katherine (18 households) shire councils. The remaining 11 households were distributed across other parts of the Northern Territory. In total, 214 households were classified as remote, and 46 households were classified as urban. Households in East Arnhem, Victoria Daly, Tiwi Islands and Katherine shire councils were predominately remote, while those in Darwin and other parts of the Northern Territory were predominately urban.

Overall, households ranged in size from one person to 23 people, with a median size of six people. Remote households were typically larger, with a median size of seven people (range 1 to 23 people) compared to a median size of four people for urban households (range one to 11 people) (Fig. 3). When stratified by shire councils, Victoria Daly had the highest median size of eight (range one to 23 people) followed by East Arnhem with a median size of 7.5 (range one to 17 people). Darwin shire council had the lowest median size of three (range one to 11 people).

The median proportion of household members who were adult in remote areas (67%, IQR 55–83%) was less than urban areas (78%, IQR 50–100%). In contrast, the median proportion of school-aged children in a household in remote areas was higher (20%, IQR 0–38%) than urban areas (0%, IQR 0–29%). However, the median proportions of pre-school aged children were almost equal in both remote and urban which are 0% (IQR 0–14%) and 0%(IQR 0–18%) respectively. The median proportion of male were equal (50%) in both remote and urban areas.

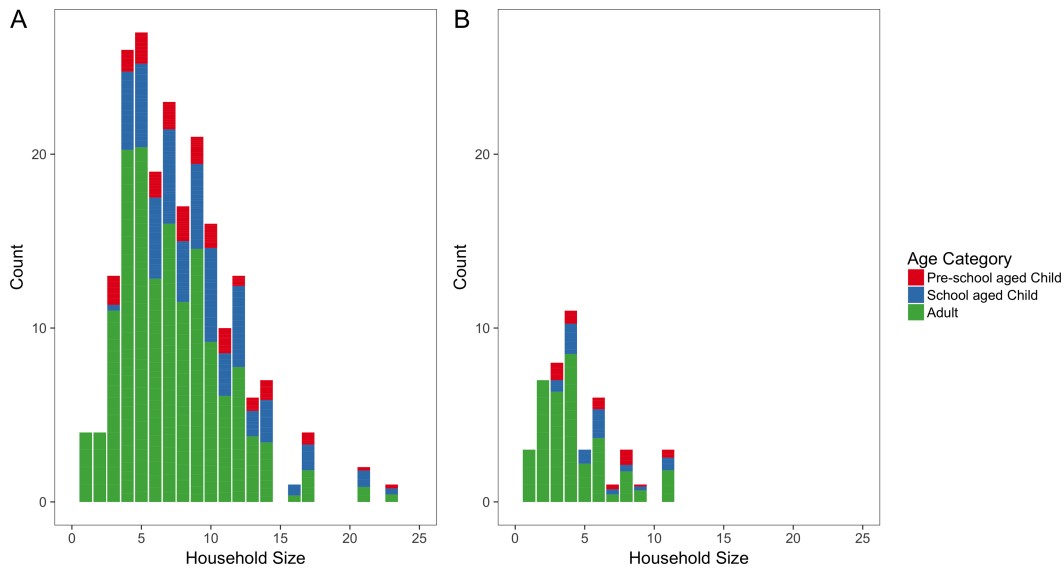

**Figure 3  Household size distributions.** Number of people per household for remote (A) and urban (B) households. Each bar is coloured according to the mean proportion of household members who are adults (blue), school aged children (green) and pre-school aged children (red).

The mean number of people per room was 2.8 (range one to six) in remote areas and was 2.4 (range one to six) in urban areas. When this is stratified by shire councils, the mean number of people per room was 3.1 in Victoria Daly which was the highest followed by East Arnhem with 2.7 and both having a range from one to six occupants. Katherine and Tiwi Islands shire councils had 2.6 and 2.3 respectively. Darwin shire council had the lowest mean number of people per room which was 2.2 with a range of one to four occupants. Figure 4 illustrates how occupancy rates vary with the number of occupied rooms. The highest room occupancy rates (5–6 people per room) tended to occur in remote households with fewer occupied rooms (one or two rooms).

## Household contact matrices

Figure 5 shows household contact matrices for remote and urban Indigenous households. The colour gradient and numerical values indicate the mean level of contact for that age category pair per household. Household contact matrices stratified by shire councils are included as Fig. S2. Contact matrices shown in Fig. 5 were calculated using a *room factor* of 1; that is, no additional weighting for individuals sharing the same room. The effect of weighting by rooms on contact matrices is shown in Fig. S3. Increasing the weighting attributed to sharing a room increases the proportion of contacts involving school aged and pre-school aged children, relative to that occurring among adults.

For comparison, we also generated a household contact matrix derived from data collected in two local government areas (LGAs) of Melbourne, Boroondara and Hume. Data were collected using a computer assisted telephone interview method. The household size was determined by the number of members living in the house. Figure 6 shows the household contact matrix created by aggregating the households in this data set. The
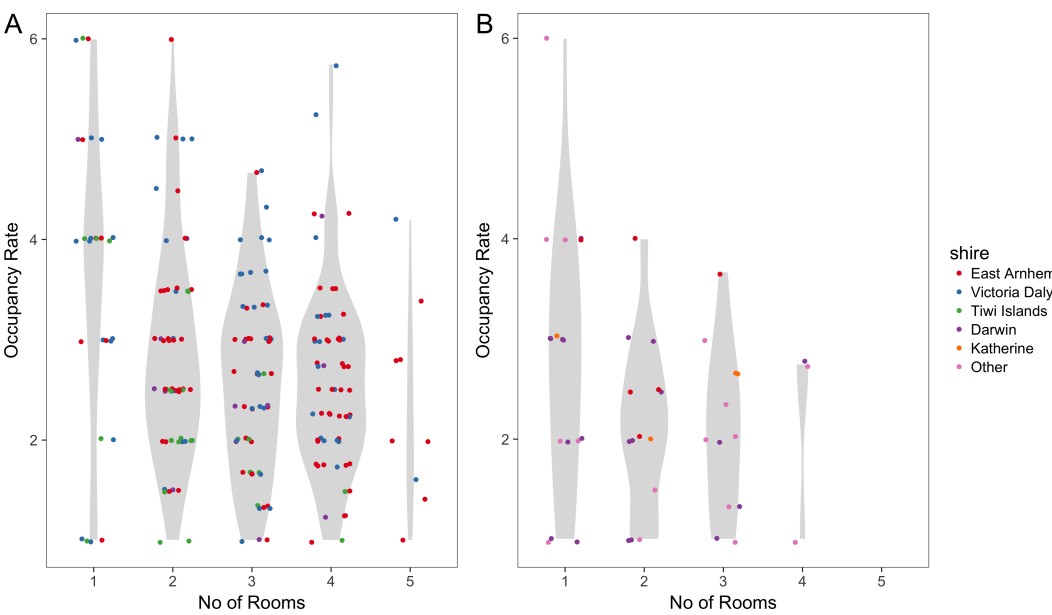

**Figure 4** **Room occupancy rates.** Dots (jittered) show the mean number of people per room, stratified by number of occupied rooms, for remote (A) and urban (B) households. The Violin plots in grey show the probability density of the data.

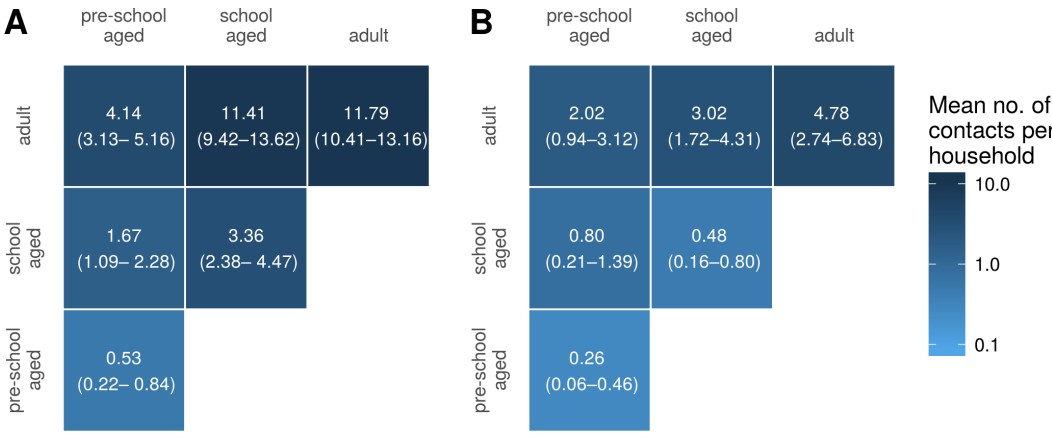

**Figure 5** **Household contact matrices—Indigenous remote and Indigenous urban.** Mean number of contacts between each age category in households in remote (A) and urban (B) communities. 95% confidence intervals estimated with a nonparametric bootstrap method are indicated in brackets.

average level of household contact (as reflected by these data sets) is an order of magnitude greater in Northern Territory houses than in Melbourne houses. These differences vary by age: while the average number of contacts among adult household members increases by a factor of approximately four, the increase among school aged children is 15–20-fold and that of pre-school aged children by 25-fold.

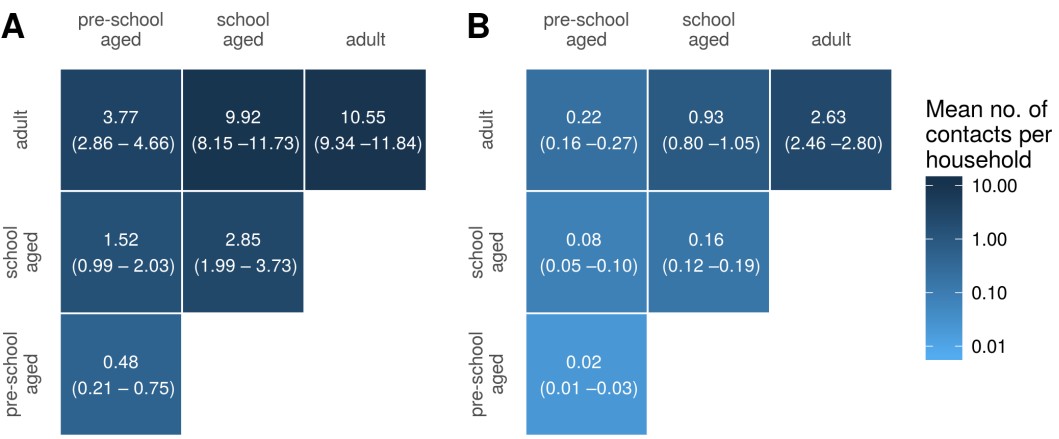

**Figure 6 Household contact matrices—Indigenous and non-Indigenous.** Mean number of contacts between each age category in households in the Northern Territory (A; with remote and urban communities combined) and in Melbourne (B). 95% confidence intervals estimated with a nonparametric bootstrap method are indicated in brackets.

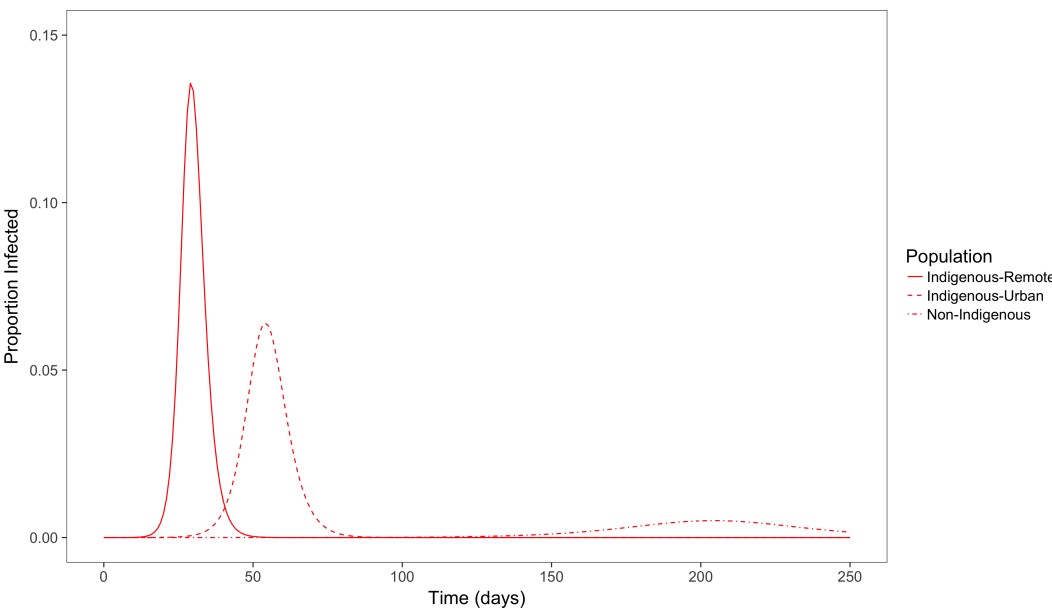

**Figure 7 Simulated outbreaks of an influenza-like illness.** Proportion of population infected over time in populations with demographic and contact characteristics calibrated to remote and urban indigenous populations and a non-Indigenous urban population.

## Outbreak simulations

Figure 7 shows the simulation outcome for the population in the infected state using a simple deterministic SEIR model.

With population and contact characteristics calibrated to an urban non-Indigenous population, the peak of the outbreak occurs around day 200 with a peak prevalence of

less than 1%. In comparison, in an Indigenous remote community the peak occurs more quickly around the 30th day with a peak prevalence of 14%. In an Indigenous urban community, time taken for the peak infectious period is also early (around 50 days) compared to non-Indigenous population, and with a peak prevalence of 6%. The total population affected by this influenza like illness for Indigenous remote, Indigenous urban and non-Indigenous urban populations are 90%, 75% and 25% respectively. The basic reproduction number ($R_0$) for Indigenous remote, Indigenous urban and non-Indigenous urban populations were 5.5, 2.6 and 1.5 respectively. This clearly demonstrates that the level of contact in households and communities for an influenza like illness affects the peak outbreak time and overall affected size in the three different populations.

## DISCUSSION

Lack of data on the household structure of Indigenous communities impacts the prediction and modelling of infectious diseases in these areas. In order to rapidly and accurately collect household structure data in a culturally appropriate way among the Indigenous communities, a simple magnetic board method was developed. Households in Indigenous communities are observed to be crowded with large household sizes and higher room occupancy rates. Remote Indigenous communities have much higher household sizes compared to urban Indigenous communities. In this study, we show that differences in household structure and household crowding have a clear implication for the transmission dynamics of infectious diseases and contribute to the heavy burden of infectious diseases in Indigenous communities.

The impact of crowded homes and higher contact patterns on infectious disease transmission can be seen in the outcome of the simulated outbreak for an infectious disease like influenza. When the other parameters are set to be equal among the populations, the difference in contact patterns shows that among Indigenous communities, outbreaks occur sooner, have a greater peak prevalence and larger final attack rate.

The methodology described is able to capture detailed data on household occupancy in a simple and robust fashion. The data collected represents a "middle way" between the extensive but comparatively coarse-grained data collected by the national census and limited but extremely detailed data collected by small-scale demographic studies (*Morphy, 2006*; *Morphy, 2007*).

The analysis of these data are subject to some limitations. Data collected may represent a somewhat biased sample due to the nature of recruitment. All households sampled will, as a consequence of the ABC study design, contain at least one member who is approximately 25 years old.

The simplicity of the data collection method imposed some limitations on the granularity of the collected data. In particular, the allocation of household members to only three age categories limits the resolution of the age-structured contact matrices that can be derived. It is worth noting, however, that the age categories chosen are typically taken to be epidemiologically significant, due to the different opportunities for mixing that these groups tend to have.

A range of methods have been used to collect contact data, including contact diaries, wearable proximity sensors and web based surveys (*Stehle et al., 2011*; *Fournet & Barrat, 2014*; *Smieszek et al., 2016*; *Smieszek et al., 2014*). Unlike some of these other methods, the magnetic board method does not collect detailed information on contact patterns such as whether an actual contact has been made among the household members, whether it is physical or non-physical, the intensity of the contact and contact happening outside the household. However, it does quantify the nature and extent of household composition in these communities. Applying these other methods in remote communities poses significant logistic challenges, and we have therefore chosen to focus on household contacts in the first instance. Contacts made in the household represent only a subset of contacts relevant to disease transmission; however, the duration and intensity of contacts occurring in households means that these are likely to play a particularly important role (*Smieszek, 2009*). As the next step, we are currently exploring approaches to collecting information on overall contact patterns in remote communities.

The question used to determine the household size in the non-Indigenous populations through a telephone survey was "How many people usually live in your household?" as opposed to "Who slept in your household last night?" in the magnetic board method. However, occupancy of Indigenous households is known to be fluid, with considerable movement of individuals among households both within and between communities (*Prout, 2008*). The current data set provides a single snapshot of household occupancy, but no way of determining how this state of occupancy may change over time. The data collection methods used, however, are well-suited to such a longitudinal study.

It is important to note that our model only focuses on the difference in contact patterns, and does not include all factors relevant to disease transmission. These factors may include the strength of contact between individuals, duration and distance of contact, difference in immunity levels and infectiousness among different age groups, pathogen levels, and the effect of vaccination (*House & Keeling, 2009*; *Smieszek, 2009*; *Rea et al., 2007*). The model also assumes that the effective contact per day depends on the number of other household members, but as the size of the household increases, the intensity of contact may differ among individuals. Certain practices among the remote Indigenous communities such as co-sleeping with infants (*Panaretto et al., 2002*), hygiene levels (*McDonald et al., 2008*) and ventilation (*Prabhu et al., 2013*) would also affect the probability of transmission of an infectious disease. These factors are difficult to quantify, but through introducing room level weights, the risk of prolonged and intense contact is captured to some extent. Although not included in our simple simulation model, given that children have been found to be more infectious than others, (*Ghani et al., 2010*; *Viboud et al., 2004*) age-specific infectiousness could be incorporated in a relatively straightforward fashion into disease models alongside the age-specific contact rates reported here.

By quantifying the extent to which Indigenous households are large and over-crowded, there is a better understanding of the extent to which model parameters estimated from non-Indigenous populations will underestimate the size and speed of outbreaks (and disease burden) when modelling Indigenous populations. This gives insight into making decisions on intervention options such as the possibility of developing vaccines during the

shorter period or allocating resources and creating awareness of communicable diseases and ways of transmission in such settings.

In the future, when conducting similar studies, a more fine-grained age structure will be useful in further understanding the contact patterns among different age groups. Currently we classified household members as only adult, school aged child and pre-school aged child. Categorizing household members into 5-year age groups would provide a more detailed picture of contact patterns and disease transmission. Also, combining the simple methodology described above with the use of mobile digital technology such as a smartphone or iPad application may enable richer data to be collected without compromising the intuitive nature of the method, and also remove the need for subsequent manual entry of data. Such advances would facilitate longitudinal but frequent sampling of households to provide a more dynamic picture of population flux within households.

## ACKNOWLEDGEMENTS

The research on which this paper is based was conducted as part of the Life Course Program. We thank the dedicated Life Course research team who traced participants and collected the data. We especially thank the young adults belonging to the Aboriginal Birth Cohort and their families and community for their co-operation and support and all the individuals who helped in the urban and remote locations. We wish to acknowledge the late Dr. Sue Sayers, founder of the ABC study.

### Funding

The project was supported by an Australian National Health and Medical Research Council (NHMRC) project grant (#1098319). Steven Y.C. Tong is a NHMRC Career Development Fellow (#1065736). Data collection for the ABC study, Gurmeet Singh and Belinda Davison are supported by the NHMRC (#1046391). Jodie McVernon is supported by NHMRC Principal Research Fellowship (#1117140). Andrew Robinson is supported by the Centre of Excellence for Biosecurity Risk Analysis. Thiripura Vino and Michael John Lydeamore are supported by an Australian Government Research Training Program Scholarship. The funders had no role in study design, data collection and analysis, decision to publish, or preparation of the manuscript.

### Grant Disclosures

The following grant information was disclosed by the authors:
Australian National Health and Medical Research Council (NHMRC): #1098319.
NHMRC Career Development Fellow: #1065736.
NHMRC: #1046391.
NHMRC Principal Research Fellowship: #1117140.
Centre of Excellence for Biosecurity Risk Analysis.
Australian Government Research Training Program Scholarship.

## Competing Interests

Steven Y.C. Tong is an Academic Editor for PeerJ.

## Author Contributions

- Thiripura Vino and Nicholas Geard conceived and designed the experiments, performed the experiments, analyzed the data, contributed reagents/materials/analysis tools, wrote the paper, prepared figures and/or tables, reviewed drafts of the paper.
- Gurmeet R. Singh conceived and designed the experiments, contributed reagents/materials/analysis tools, reviewed drafts of the paper.
- Belinda Davison contributed reagents/materials/analysis tools, reviewed drafts of the paper.
- Patricia T. Campbell and Michael J. Lydeamore performed the experiments, analyzed the data, contributed reagents/materials/analysis tools, reviewed drafts of the paper.
- Andrew Robinson and Jodie McVernon reviewed drafts of the paper.
- Steven Y.C. Tong conceived and designed the experiments, wrote the paper, reviewed drafts of the paper.

## Human Ethics

The following information was supplied relating to ethical approvals (i.e., approving body and any reference numbers):

The Human Research Ethics Committee of NT Department of Health and Menzies School of Health Research approved this study (ABC Reference no. 2013–2022).

## Data Availability

The de-identified raw data has been provided as a Supplemental File.

## Supplemental Information

Supplemental information for this article can be found online at http://dx.doi.org/10.7717/peerj.3958#supplemental-information.

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
