# Peer review of "Indigenous Australian household structure: a simple data collection tool and implications for close contact transmission of communicable diseases"

_PeerJ, doi:10.7717/peerj.3958_

## Round 0.1 · original submission · Major Revisions

· Academic Editor

Major Revisions

Please check comments on design issue, including the one about the size of respondents as noted by the reviewer 2. Both reviewers kindly suggested useful references. I'd like you to check all these and think whether they fall into the area of your submitted study, in line with your Discussion.

[# Staff Note: It is PeerJ policy that additional references suggested during the peer-review process should only be included if the authors are in agreement that they are relevant and useful #]

·

Basic reporting

Basic reporting is good, with clear presentation of the work. No concerns about this.

There is one typo in equation 2 (page 4): the minus sign in front of the lambda should not be there.

Perhaps there could have been a bit more of a discussion of the literature relating to measuring contact patterns relevant to infectious diseases and the potential limitations of relying on the number of household contacts alone. Some possible reference are

Ogunjimi B et al Mathematical Biosciences 2009
Xiao X et al Epidemics 2016
Melegaro et al Epidemics 2011
Zagheni et al AJE 2008
Stehle et al PLoS One 2011
Fournet et al PLoS One 2014

One limitation of relying on contact data alone is children have consistently been found to more infectious than others (for example, see Ghani et al 2010 paper cited, but several others as well such as Viboud et al Br J Gen Prac 2004).

Would be nice for the reader if Fig 3 could use colours corresponding to those chosen in Fig 2.

Minor point: "data are" is usually preferred to "data is" (though I note the Peerj review website uses "data is" so I'm probably fighting a losing battle on this one).

Experimental design

I have no concerns about the experimental design. I believe this is relevant, original, meaningful research with appropriate methods that clearly fills a knowledge gap.

The outbreak simulations are probably the weakest part of this (it seems a bit odd to collect good household data but then not look at a household model). However, this is not an essential part of the paper - this is more just to illustrate the point that contact patterns can be important. These simulations do not represent a novel finding and could easily be dropped without affecting the quality of the paper. If these stay in I think at least worth discussing limitations of this model (which can be contrasted with a household model), + also reporting R0 values for the three populations considered. Perhaps also worth mentioning that different methodology was used to collect data from the Melbourne site, so contact matrices are not necessarily comparing exactly like with like.

Validity of the findings

I have no reason to doubt the validity of the findings and the conclusions, though I think there could be more of a discussion of the relevance of the household data alone for informing transmission models. First because the number of effective contacts made by a single individual with household members may not necessarily scale linearly with the number of other household members, and secondly because many other factors (other than numbers of people per household and per room) may be important determinants for household transmission risk. Most importantly age, but other factors such as ventilation/humidity + hygiene practices may also be important.

Comments for the author

-

·

Basic reporting

The manuscript is well-written (structure and language).

From my point of view, the analysis of the data should be deepened and better embedded in discussions around validity / reliability of measurements (see section 3) and relevant literature should be cited. I will suggest literature in section 3.

Experimental design

I have a few methodological questions:

1) How comes that the authors had more respondents for the magnetic board than for the questionnaire? I thought that questionnaire participation was mandatory and magentic board was not; hence the magnetic board respondents should be a subset of the questionnaire respondents? (ll. 94/95)

2) The authors state that the magnetic board was separated into four rooms. What happened if the respondent's house had more or less than four rooms? (l. 108)

3) How exactly was the weighting done (ll. 150/151)? Maybe an equation would help?

Validity of the findings

1) The authors introduced a new method to measure contacts within a household. At least some data were measured with two methods (questionnaire and magnetic board). From my point of view, a new methods should be thoroughly assessed re how reliably data are collected. What are potential biases? Is the underlying contact definition the same as in other methods? If not, what are the potential implications of that?

There is a huge body of literature on contact measurement methodology. Some paper that might be helpful to deepen this analyses are the following (in reverse-chronological order):

BMC Infect Dis 2016, 16: 341
PLOS ONE 2015, 10: e136497
BMC Infect Dis 2014, 14: 136
Epidemiol Infect 2012, 140: 744-52
PLOS ONE 2012, 7: e37893
Proc Biol Sci 2011, 278: 1467-75

2) Since the numbers of respondents are at the lower end, it would be advisable to calculate confidence intervals when creating contact matrices for specific strata to assess whether apparent differences are statistically significant. This could be done with bootstrapping.

3) Since the authors have introduced weights for being in the same room, they also might want to discuss this in light of the literature around contact duration / intensity, e.g.

Biostatistics 2014, 15: 470-83
Epidemics 2011, 3: 143-51
R Stat Soc C 2010, 59: 255-77
Theor Biol Med Model 2009, 6:25
Epidemiol Infect 2007, 135: 914-21

Comments for the author

- ll. 55/56: what kind of skin infections? Impetigo?
- l. 81: what is meant with "Top End"? The northern part?
- l. 153 what is a "shire council"?

---

## Round 0.2 · Minor Revisions

· Academic Editor

Minor Revisions

Please address few remaining minor comments

·

Basic reporting

Changes are good. Just a few minor typos to be fixed (line numbers are from marked up version):

line 99 punctuation: ”. The questionnaire asked the question ?Who slept in the house last night?”


line 221 + 260 "NT" used inconsistently. I think better to spell out Northern Territory each time as abbreviation isn't used often.

line 252 "proportion of contact" ->"proportion of contacts"
line 276 "population" -> "populations"
line 277 "household" ->"households"
line 278 remove comma
line 290 "contribute" - >"contributes"
line 309 & 332 comma needed: ...., however, ...


Also, no mention of methodology used to calculate the R0s which have been added in the results section in this revision. Presumably dominant eigen-value of next generation matrix, but worth saying and including the appropriate reference (Diekmann + Heesterbeek).

Experimental design

no comment

Validity of the findings

no comment

---

## Round 0.3 · accepted · Accept

· Academic Editor

Accept

I can tell that all the comments have been properly addressed. Thank you very much for your meticulous work.